# Translating concerns into action: a detailed qualitative evaluation of an interdisciplinary intervention on medical wards

Samuel Pannick,[1] Stephanie Archer,[1] Maximillian J Johnston,[1] Iain Beveridge,[2] Susannah Jane Long,[1,3] Thanos Athanasiou,[4] Nick Sevdalis[5]

[1]NIHR Imperial Patient Safety Translational Research Centre, Imperial College, London, UK
[2]Department of Medicine, West Middlesex University Hospital NHS Trust, London, UK
[3]Imperial College Healthcare NHS Trust, London, UK
[4]Department of Surgery and Cancer, Imperial College, London, UK
[5]Centre for Implementation Science, King's College, London, UK

**Correspondence to**
Dr Samuel Pannick;
s.pannick@imperial.ac.uk

## ABSTRACT

**Objectives** To understand how frontline reports of day-to-day care failings might be better translated into improvement.

**Design** Qualitative evaluation of an interdisciplinary team intervention capitalising on the frontline experience of care delivery. Prospective clinical team surveillance (PCTS) involved structured interdisciplinary briefings to capture challenges in care delivery, facilitated organisational escalation of the issues they identified, and feedback. Eighteen months of ethnography and two focus groups were conducted with staff taking part in a trial of PCTS.

**Results** PCTS fostered psychological safety—a confidence that the team would not embarrass or punish those who speak up. This was complemented by a hard edge of accountability, whereby team members would regulate their own behaviour in anticipation of future briefings. Frontline concerns were triaged to managers, or resolved autonomously by ward teams, reversing what had been well-established normalisations of deviance. Junior clinicians found a degree of catharsis in airing their concerns, and their teams became more proactive in addressing improvement opportunities. PCTS generated tangible organisational changes, and enabled managers to make a convincing case for investment. However, briefings were constrained by the need to preserve professional credibility, and staff found some comfort in avoiding accountability . At higher organisational levels, frontline concerns were subject to competition with other priorities, and their resolution was limited by the scale of the challenges they described.

**Conclusions** Prospective safety strategies relying on staff-volunteered data produce acceptable, negotiated accounts, subject to the many interdisciplinary tensions that characterise ward work. Nonetheless, these strategies give managers access to the realities of frontline cares, and support frontline staff to make incremental changes in their daily work. These are goals for learning healthcare organisations.

**Trial registration** ISRCTN 34806867.

## INTRODUCTION

Around 10% of hospitalised patients suffer preventable harms.[1 2] Many are the result of persistent organisational failings, either deliberately ignored, or to which the organisation has become blind.[3] Frontline staff have a unique insight into these failings in day-to-day care, identifying safety and quality concerns in the course of their routine work.[4–7] Their concerns—revealing uncomfortable institutional fallibilities—are not reflected in high-level organisational metrics. Rather, frontline insights into everyday processes are 'soft data', adding a nuanced understanding that can build more holistic improvement strategies.[8]

Senior healthcare leaders acknowledge the importance of this frontline knowledge—yet those same leaders are loath to spend time and resources pursuing staff concerns that might ultimately prove to be trivial.[9] Few safety campaigns focus on the tribulations of frontline workers.[10 11] Instead, staff typically devise their own workarounds for operational problems, producing temporary fixes. This 'first-order' problem solving is effective in the moment—but it does little to prevent each problem's recurrence, and organisational processes go unchanged.[12–14]

## Strengths and limitations of this study

► Qualitative evaluations of complex interventions are recommended by the Medical Research Council.
► This study identified the mechanisms to be replicated if further prospective clinical team surveillance implementation efforts are to prove successful.
► The evaluation highlighted tensions when trying to capture the frontline experience of care delivery; previous reports had characterised frontline data collection as a more objective exercise.
► Selection bias may reduce the internal validity of our findings: there was disproportionate access to staff who engaged with the intervention.
► As a single-site study, the generalisability of the results may be limited.

'Prospective clinical surveillance' is one mechanism for improving care delivery from a frontline perspective. Embedded observers (or visiting facilitators) work with frontline staff to record their experiences of care delivery and its consequences, with a structure to support data capture.[2 15 16] Documenting potentially flawed care processes and adverse events, prospective clinical surveillance can produce detailed, unit-level performance assessments in near-real time.[17] It is 'prospective' in two respects. First, it proactively seeks out staff concerns, rather than waiting for staff to volunteer them. Second, it can identify care deficiencies that have not yet led to patient harm. However, its effectiveness as an improvement strategy is unclear. Whether prospective clinical surveillance translates staff concerns into tangible action, and how exactly it might do so, have not previously been explored.

Here, we investigate the qualitative impact of prospective clinical *team* surveillance (PCTS), a novel extension of this technique. PCTS had three components: (i) structured interdisciplinary briefings to capture frontline clinical and administrative challenges; (ii) facilitated organisational escalation of the issues they identified and (iii) feedback. PCTS was evaluated in a cluster controlled trial on medical wards, involving seven interdisciplinary teams from two London (UK) hospitals.[18] High fidelity PCTS reduced excess length of stay; it also improved teamwork and safety climates, and incident reporting.[19] Lower fidelity implementation may have had a detrimental impact on team performance. In this study, we seek to describe the intervention's effects as understood and enacted by participating staff.

## METHODS

### Intervention
The intervention is described in detail in the published protocol.[18] In brief, structured briefings reviewed the events of the previous shift, identifying clinical and administrative challenges and concluding with a plan to resolve or escalate those concerns. A structured pro forma (Hospital Event Analysis Describing Significant Unanticipated Problems (HEADS-UP)) focused on the problems most commonly identified on medical wards.[20] After a short period of supervision, ward teams led their own briefings autonomously. A single facilitator helped teams advance the issues raised in their briefings, and provided follow-up and feedback to stakeholders throughout the organisation. Briefings were known locally as 'HEADS-UP briefings'. The autonomous team briefings were intended to be a sustainable local alternative to the facilitator-led data capture described in previous reports.[2 16]

### Process evaluation
This study was grounded in the Medical Research Council framework for process evaluations of complex interventions.[21] Qualitative methods identify the complex causal pathways and unexpected mechanisms of impact by which these interventions produce change. Ethnography and focus groups are recommended for this type of investigation:[22] they are used here to evaluate the intervention at the hospital site most heavily involved in the primary study.

### Embedded research and auto-ethnography
The programme lead was an embedded researcher at the institution contributing six of the seven study wards. This position is defined as 'work(ing) inside (a) host organisation as (a) member of staff, while also maintaining an affiliation with an academic institution… conduct(ing) research studies… which respond to the needs of the organisation, and accord with its unique context and culture'.[23] Relationships between staff and the researcher are important: the researcher is seen both to be part of the team and contributing to the research capacity of the host organisation.[24] Fieldnotes, typed contemporaneously as soon as was practicable, recorded personal exchanges with staff, changes arising from PCTS, governance proceedings and reflections on the broader context. These 'auto-ethnographic' insights have been widely used in organisational case study research, providing broad accounts of culture and practice.[25–27]

### Focus groups
Semi-structured focus groups were undertaken in July 2015. The topics of interest were specific attitudes, feelings and beliefs that would best be revealed through social interactions between group participants.[28] The topic guide explored PCTS as a tool, as well as the implementation process that introduced it. To orientate them, participants were asked to discuss existing systems for identifying team concerns and improving patient care. They then reflected on their experiences of using PCTS. Questions and follow-up probes scrutinised team-wide involvement in PCTS; whether it affected perceptions of the ward's safety and quality; how it was used in staff interactions and whether changes had been made as a result. Topic guide questions were reviewed and piloted within the research team, and with researchers in healthcare quality and safety who had not been involved with the study previously. The focus groups were advertised in clinical areas, by email and in person; all ward staff were invited to attend. Certain staff groups were purposively targeted to ensure that key stakeholders (junior doctors, senior nurses and service managers) were represented. Focus groups lasted approximately 2 hours, facilitated by experienced qualitative researchers (NS, MJJ, SA; also Dr Louise Hull (Research Fellow) and Ms Tayana Soukup (Research Assistant)), using the topic guide flexibly in view of the different roles and experiences within the two groups. Focus group discussions were digitally recorded in the hospital's medical education centre, and then professionally transcribed.

Focus group transcripts and fieldnotes were managed with NVivo software (QSR International, Doncaster, Australia). Two researchers (SP, SA) read and reread the

source material, adopting an inductive (theory-generating) approach.[29] This type of analysis is a flexible research tool, generating a rich and detailed account of a complex data set. It can be applied to focus groups as well as other qualitative data,[30–32] allowing thematic integration in a single analysis. Each researcher coded transcripts individually, generating an initial coding frame, which was then discussed and refined. The transcripts were coded again, before a group of higher order themes was agreed. Field notes were searched for evidence to support or contradict the evidence from the focus groups, and for themes not covered independently by focus group discussions. Themes were inspected against the broader literature to inform their interpretation. The study was approved by research and development authorities at participating institutions as a quality improvement programme. Focus group participants gave their signed consent. The primary study was registered with the ISRCTN registry (ISRCTN34806867).

## RESULTS

Fifteen staff participated in the focus groups: three junior doctors (foundation doctors, in their first 2 years of medical practice), eight senior ward nurses and four non-clinical managers. Participants represented each of the clinical divisions that had taken part in the study at this hospital (care of the elderly, acute medicine, respiratory medicine and gastroenterology). Focus group data were supplemented by 44 pages of auto-ethnographic field-notes, documenting the 18-month period in which links were established with clinicians; clinical and non-clinical managers and senior hospital leaders.

Our findings are organised around four themes. First, we describe the shared team beliefs and ethos established by PCTS. Second, we discuss how teams used the programme to triage ward problems more effectively. Third, we identify how individuals' practice changed as a result of their participation. Finally, we note how PCTS altered team and organisational approaches to improvement, and the tangible changes that came about.

### PCTS established psychological safety, with a hard edge of accountability

By the end of the study period, all staff groups had participated in—and led—HEADS-UP briefings:

> (Consultants) contribute as much as anyone else (…) so their voice is being heard, because they suffer from the same frustrations as all of us. *(Service manager)*
> The physio(therapists) or the OTs (occupational therapists) are also flagging things up, that's how we found it useful. (…) The discharges that did not happen, anyone (who) deteriorated (…) we all learn from what went wrong. *(Senior nurse)*

Perhaps because different staff groups were perceived as partners in them, the briefings formed a psychologically safe environment in which problems could be discussed openly, without fear of retribution. Team psychological safety does not imply undue permissiveness or unrelenting cheeriness, but a confidence that the team will not embarrass or punish someone for speaking up:

> It was more of a constructive exercise, where everyone is then wanting the same outcome, so it wasn't like, "Oh this didn't happen, I am angry at you", it was more like we are all actually in the same team (…) You all have the same end point and I think that is why it is quite safe to bring it up, because it is not confrontational, and it is not personal against someone, it is just what needs to be done. *(Foundation doctor)*

The style and timeliness of PCTS feedback also contributed to a sense that this was a non-judgemental forum for team learning:

> There were a few teaching sessions where (…) we went through things like ECGs (electrocardiographs) that were missed, and then you had a collective opportunity to think (…) It was a very good plot for learning, saying this happened and let's do some learning and try and prevent it happening again. That is quite helpful. *(Foundation doctor)*

This psychological safety was not limitless: there were still boundaries around what could be discussed. Although the briefings did prompt teams to consider positive notes from the previous shift, they rarely did so. Reflections often highlighted the overall management of patient flow, rather than specific actions or diagnostic processes that others could emulate. This reflected a reluctance to delve too deeply into any one team member's performance. Professional identities were protected by assigning problems to other departments, acknowledging procedural complications as foreseeable, or linking delays to understaffing. In this way, perceived threats to professional identity were largely deflected by a projection of blame onto other groups, or attributed to circumstances beyond the team's control.

More introspective teams, with senior support and increasing psychological safety, did record issues like diagnostic errors, in which they had played a more overt role. However, even in those teams—where the briefings were implemented with greatest fidelity—there could be disputes as to whether certain problems were really problems at all:

> The senior nurse disagrees with medical teams' perceptions that the site moves (moving patients from one ward to another) were inappropriate. She thinks doctors expect patients to remain on the acute medical unit even when they are stable… She felt an inadequate handover perpetuated their concerns.' *(Fieldnotes)*

Thus, while psychological safety was an important factor in helping staff speak up about their concerns, certain topics remained off-limits, and psychological safety did not guarantee agreement about what were reasonable concerns. These are important limitations:

I think it's more (than) just raising the issue. (…) Give me a more detailed timeframe, resolution, how do you resolve it? Are you going to get back to me because it's important? And what will I expect? *(Senior nurse)*

Where they were most effective, the briefings maintained a 'hard edge': team members regulated their own behaviour, knowing that their actions (or inactions) might be flagged up at the next briefing, or in feedback. Junior doctors and nurses were conscious of this internalised discipline (panopticism), acknowledging how it changed their own behaviour and how they could use it to their advantage. A self-reinforcing virtuous cycle emerged, with participation, accountability and feedback:

The group was more aware that if you perhaps missed something like that, it may be brought up later at a HEADS-UP meeting. (…) It made you more accountable. *(Foundation doctor)*

It certainly works for the VTEs [venous thromboembolism assessments, completed by junior doctors, often at the behest of nursing staff]. There are days that we will have eight or 10 VTEs (to do), so if I (…) give the list to the doctor, the following day (…) there will be one or two, because they know (…) it will be mentioned again, with the consultant. *(Senior nurse)*

You get more feedback, so you raise concern(s), and then maybe the next day the same issue would crop up, and you actually then get feedback on what they are doing about it, or who they have escalated to. *(Foundation doctor)*

As we will now describe, airing and recording concerns in this environment helped triage those problems, directing them back to frontline teams or onwards to their managers.

### Rapid resolution and meaningful managerial follow-up

To explain their perceptions of PCTS, focus group participants drew comparisons with the existing processes for identifying and resolving ward-level problems. The hospital's online incident reporting system was considered the formal mechanism by which problems were recorded. The system was not easy to use for this purpose, and there were few attempts to persistently report recurrent problems. As a result, commonplace issues were no longer even considered abnormal—a materialisation of normalised deviance, where staff become so desensitised to deviant practice that it no longer feels wrong:

Some things staff have got so used to that they don't (report) it. So it's just become common practice. *(Clinical risk manager)*

Managers too were frustrated by the incident reporting system. Reports did not necessarily illuminate what was happening at the frontline. Moreover, long delays in processing those incident reports meant that even

relatively simple problems were not reviewed for many weeks. As a result, reports that described patient harm were prioritised: where no harm had occurred, little was done. Service managers were aware that they paid attention to only the most serious incidents:

You could go down there (to the ward) and you could see people working very, very hard and you could also sense a lot of frustrations (…) but you weren't getting to what those frustrations were, and what those issues were. *(Service manager)*

It has got to that point of being a SI (serious incident, an adverse event with particularly grave consequences) before it is then addressed and taught and learnt from. *(Service manager)*

In contrast, PCTS helped identify a route for more rapid resolution of practical problems. It provided an acceptable mechanism for staff to log issues into which they had immediate insight. This was a materialisation of second-order problem solving, staff finding a way to bring about a more lasting solution to each issue they encountered. Where issues could not be resolved within the ward team, managers were presented with clear, actionable topics to address. Those managers were then better prepared to handle incipient risks to service quality:

(PCTS) highlighted little things, that could be sorted out quite quickly. It made sure that somebody was allocated to deal with that on the day—or did you follow that through, did that get sorted out? So everything benefits, rather than just going on and on and on (…) Things get sorted out quicker. *(Senior nurse)*

What I like about it is (that) it is instant and it is thematic very quickly, you understand what the issues are (…) Here is an opportunity to dump them (your daily issues) and suddenly a picture forms out of it (…) And not just here, but there, there, there and there (…) And it hasn't yet created a significant incident, but clearly there is risk associated with it. *(Service manager)*

Therefore, PCTS brought about faster resolution of safety and quality issues through a combination of mechanisms. First, increasing self-monitoring within ward teams helped anticipate and mitigate potential problems. Second, senior ward staff at the daily briefings were more quickly aware of issues they could resolve, without having to trawl through a backlog of outstanding incident reports. Third, facilitation drew out thematic problems across departments, bringing them to the attention of middle managers who had struggled to understand the difficulties experienced by their staff. Fourth, the clinical impact of these problems was made clearer to senior executives, who were motivated by these novel frontline narratives to pursue change:

PCTS findings informed the discussion in the Trust morbidity and mortality meeting, chaired by the

chief executive, highlighting areas which needed faster examination and transformation, and bringing genuinely new information… Many of these concerns had not been formally addressed either in incident reports or top-down initiatives… It also highlighted new targets for improvement. *(Fieldnotes)*

However, not all managers were equally enthused. Where PCTS had not provided data that appeared directly useful for them, middle managers did less to hold their service areas accountable for briefing implementation. Board members questioned how best to detect a meaningful signal among the noise of staff reporting (whether through formal incident reports or PCTS). At times, managers seemed nihilistic about the possibility of improvement, or questioned whether their frontline teams truly understood the problems they had reported:

> We have two chances of getting (this issue resolved)—fat chance and no chance! We are monitoring (this) on a daily basis but (…) the situation is far from ideal and we have escalated up the chain – not that it has helped. *(Support service manager—Fieldnotes)*
> I would like to get some feedback on (these issues) first (…) Currently they are only perception(s), and following investigation it may be that the risk changes, or the issue is solely around communication of processes already in place. *(General manager—Fieldnotes)*

Still, the combination of increased frontline efficiency and managerial involvement (at least in some areas) led to a more general sense that things had changed for the better:

> (PCTS) would make it clear what was actually deficient. And then I think things did start to change, and you did see the feedback coming through (…) So you could definitely see the evolution of it. *(Foundation doctor)*
> There (are) also the (…) day-to-day things within the team that start running better, and I think you do start seeing those, but it is harder to put a specific, "This was raised, this was done, there was an outcome." It is a more (…) general change as part of (PCTS). *(Foundation doctor)*

With a structure in place to coordinate frontline teams and their managers, individual staff found their own practice changing, with unexpected personal benefits. These will now be discussed in more detail.

### Changing individuals: reversing normalised deviance, and catharsis

PCTS prompted staff to proactively address issues they may have previously ignored. This was, in effect, a reversal of the normalised deviance that had become so ingrained:

> You wouldn't have identified that (issue) necessarily, because it is just part of your daily (…) life in the NHS, but when it is put like this (with PCTS), it is

highlighted, you have got an opportunity to really do something about it. *(Service manager)*
> This particular patient (has been) awaiting echo(cardiography) for the last three days (…) (I) would now question why is it still not done (…) When I wasn't doing (the briefings), I wasn't seeing the importance (…) (Now) I tell the Matron, or I tell my Bed Manager […] or I go there myself (…) I go to the department and ask them, "This is what's happening, this is what I need." *(Senior nurse)*

Having made those problems visible once again, the information shared in the HEADS-UP briefings would be used to actively coordinate plans and decisions for the next shift. Aware of pressures on their colleagues, and cognisant that problems were occurring repeatedly, individuals would change their routines:

> I tell the physio(therapist) and the OT (occupational therapist): "These are your priorities for today, these are your priorities for tomorrow." So they already have a plan (…) Like I said, it gives you the structure that you need. *(Senior nurse)*
> The group as a whole became more "present" to (what) happens maybe a couple of times in a week (…) For example if the nursing staff are short (…) just having that in your mind and thinking "No, we need to get all these blood tests done now", you would appreciate how the service could be best run. I think that made it more efficient. *(Foundation doctor)*

The improved information sharing, interdisciplinary coordination and efforts to evenly distribute workload produced a more supportive team climate. In this atmosphere, the briefings could become unexpectedly cathartic, giving a voice to junior clinicians to air their personal worries, as well as their clinical concerns:

> You felt like you had got a lot of your chest by doing it (…) The next day you could say, "These are the things that went wrong yesterday", and now I can clear that and start again. *(Foundation doctor)*
> (It) makes you feel more heard, because otherwise you are just venting your concern, but you don't know if it has been actioned in any way. And then you can provide the objective evidence to move forward, to change something. *(Foundation doctor)*

The combination of changes in team ethos, triage and individual practice ultimately provoked a new approach to improvement, and a degree of organisational development.

### Changing teams and the organisation

Contrary to some leaders' fears, PCTS did not reduce the volume of formal incident reports. Rather, it provoked new reports, and altered how those reports were addressed. Yet the translation of briefing-recorded incidents into online reports was imperfect, not least because there was no agreed list of high-priority events to be recorded:

---

**Box 1    Examples of service investment and organisational change arising from PCTS**

Cost-neutral re-alignment of porter provision for radiology, doubling the in-patient flow for X-rays

Re-evaluation of phlebotomy service provision, ultimately generating a £20 000 investment in an expanded support service

Additional phlebotomy training for healthcare assistants, delivered internally

Additional sonographer in echocardiography

Relocation of intravenous infusion and nasogastric feeding pumps to a central equipment library

Development of electronic referrals and electronic reporting for medical specialty consultations, using existing electronic health record

£5000 investment in ketone meters for management of diabetic ketoacidosis.

---

Antibiotics had not been given (…) The junior doctors had not seen the procedure report specifying they were needed. No incident report was done, even though (the registrar) specifically asked if one should be completed. *(Fieldnotes)*

When incidents were reported, PCTS briefings allowed incident investigators to start to collect relevant information. The problems that had provoked the incident were typically captured by the briefing, giving the investigator (often the nurse in charge) a head start in understanding what had happened:

The people involved are already there, they are already able to explain to you, "This is what actually happened from our end." (…) It's easier for you to do your incident (investigation) report already because as the investigator you don't have to call everyone. *(Senior nurse)*

The emphasis on clinical teams' accountability also led to a subtler change in incident management. Teams took responsibility for issues that were actually within their power to change, rather than awaiting permission or hoping that the problems would be resolved elsewhere:

Very often before it gets to me, it has been dealt with and sorted out by the teams themselves (…) With HEADS-UP and (incident reporting) together, there is a different approach: (…) "We have discussed it, we have taken on the responsibility of this, we have done as much of this as we can, but here is the point where it has to be escalated and we want to make sure something is done about it, and this is our methodology for doing it." That is how it feels to me, anyway. *(Service manager)*

At multiple levels within the organisation, it was clear that this programme had revealed hidden gaps in practice. PCTS proved useful even on wards where teams were already thought to be functioning well:

There is definitely time to do it, there is definitely a need to do it, but (…) you only know that once you have done it. *(Service manager)*

I don't think we would have known, unless we had done it, that there was a gap there. *(Senior nurse)*

Similarly, in safety and governance meetings, PCTS findings highlighted new areas for improvement, many of which had not been addressed in incident reports or top-down initiatives. Those findings enabled managers to make a convincing case for investment, with numerous specific, tangible changes driven by PCTS during the study period [box 1]. These changes were described as 'quick wins' by senior managers, with clinicians agreeing that they were at least a relatively rapid organisational response:

It empowers me then as manager for that area to go forward with a business case with the evidence to say we need additional resource, this is the implications of it (…) And it works, so we have an additional (cardiac) sonographer (…) I think we saw very quickly a reduction in delays once that was sorted. *(Service manager)*

However, internal investment was not necessarily targeted specifically to participating teams. For example, expansion of the phlebotomy service (identified through PCTS as no longer meeting patient needs) primarily benefited non-participant wards, which were deemed by senior management to be more in need of support. Nor was progress guaranteed by repeated ward-level reporting. Changes to structures or processes were more likely when there was an associated financial target; another organisational incentive aligning with the proposed improvement and when clinicians and managers agreed the need for change.

Additionally, some of the concerns raised by frontline staff simply could not be addressed within the organisation. Poor staffing levels, for example, were felt keenly, but reflected a broader national challenge. Creative workarounds and mitigation plans could only go so far in addressing a fundamental shortfall in staff numbers. This was frustrating for the teams completing the briefings. PCTS also identified issues like unsafe inter-hospital transfers, whose resolution would have required close collaboration within a network of hospitals. With many other competing priorities, a local improvement effort was no guarantor of the necessary cooperative drive for change.

## DISCUSSION

PCTS established a basis of team psychological safety, married with a hard edge of accountability. Ward teams triaged recurrent problems to their managers, or resolved them autonomously, reversing normalised deviance through non-judgemental facilitation and feedback. Junior clinicians found a degree of catharsis, and teams became more proactive in addressing improvement

opportunities. The soft intelligence provided through PCTS proved effective in generating tangible organisational change—although unpredictably, dependent on the hospital's other priorities and incentives. At the ward level, briefings were limited by the need to preserve professional credibility, a lack of interdisciplinary consensus about the nature of the problems they were seeing and the relative comfort afforded by the avoidance of accountability. At higher organisational levels, frontline concerns were subject to competition with other priorities, and resolution was limited by the full scale of the challenges they described.

To our knowledge, this report is the first qualitative analysis of how this type of interdisciplinary intervention might generate improved outcomes. Previous observational studies of prospective clinical surveillance had portrayed staff as objective data recorders: how they might actively shape, frame and mitigate safety concerns was not considered. Our current study therefore offers two novel contributions. First, it identifies the mechanisms to be replicated if other PCTS implementation efforts are to prove successful. Second, it highlights the innate tensions within these efforts to capitalise on the frontline experience of care delivery.

There are multiple sources of friction in organisational learning. For example, when reporting their concerns, staff portray 'others' as threatening patient safety, rather than jeopardising their own credibility.[33 34] Disputes about appropriate care are common, not least because there are multiple interpretations of what the right course of action *should* have been.[35 36] These distinctions are lost as concerns are escalated through an organisation: rich narratives about patient safety events are progressively sanitised of their context.[34] Thus, while senior executives are encouraged to obtain 'first-hand knowledge of the system',[37] there is no single unproblematic version of the reality of frontline care, however we seek to understand it. One study of executive walkrounds (visits to the 'shop floor') suggested that these visits, whether announced or unexpected, necessarily revealed partial, biased accounts of what was going on.[38] Here too, we found that our structured programme to characterise frontline care was subject to interdisciplinary (and clinicomanagerial) disputes and tensions. Prospective clinical surveillance has been described as 'the future of measuring patient safety'.[17] We would now argue that these systems are as value-laden and subjective as any other attempt to understand frontline care delivery.

Limitations of this study include our disproportionate access to the teams and individuals who engaged with PCTS. These enthusiasts may have had a more positive view of the programme than the wider group of eligible clinicians and managers. Selection bias has hampered mixed methods evaluations of other quality improvement interventions.[39] We attempted to counteract this bias with focus group facilitators who had not been directly involved in delivering the PCTS programme, and searched for conflicting opinions in the group transcripts

and fieldnotes. Our results may also have been influenced by social desirability bias—participants giving the acceptable 'right' answers rather than revealing their true thoughts—and may not generalise fully to other settings. Nonetheless, our findings are consistent with, and build on, the existing understanding of frontline safety engagement.[40] Strengths of the study include the theoretically informed analysis of an emergent safety surveillance strategy, and dual methods allowing themes to be integrated. The embedded research model produced good access to managers as well as frontline staff: often, there is a trade-off between the two. Importantly, the study identified mechanisms by which frontline concerns can be used productively, rather than disappearing into an organisational 'black hole'[13] from which they generate no change or learning.

## CONCLUSIONS

Healthcare systems can only improve with a detailed operational understanding of how care actually takes place.[41] Prospective clinical surveillance strategies offer a route to this understanding. These strategies are not merely novel measurement tools, as they have been described[15–17], even if they do get closer to the realities of frontline care than other safety systems. They still produce negotiated, acceptable accounts, subject to the many interdisciplinary tensions that characterise ward work. Nonetheless, prospective surveillance strategies—through which system flaws are mitigated as they become apparent—facilitate continuous safety improvement. They foster improvement by making soft data intelligible to healthcare managers, and by supporting frontline staff to make incremental changes in their daily work—a goal for learning healthcare organisations.

**Acknowledgements** We are grateful to Dr Louise Hull and Ms Tayana Soukup for their help in facilitating the focus groups.

**Contributors** Study design: SP, SA, MJJ, TA, NS, SJL, IB. Study implementation and data collection: SA, SP, MJJ, SJL, IB. Analysis: SA, SP, NS. All authors contributed to, read and approved the final manuscript. Dr Pannick had full access to all of the data in the study and takes responsibility for the integrity of the data.

**Funding** This paper represents independent research supported by the National Institute for Health Research (NIHR) Imperial Patient Safety Translational Research Centre, Imperial College Healthcare Charity (Grant GG14\1022), and West Middlesex University Hospital NHS Trust. The research by NS is supported by the NIHR Collaboration for Leadership in Applied Health Research and Care South London at King's College Hospital NHS Foundation Trust. NS is a member of King's Improvement Science, which is part of the NIHR CLAHRC South London and comprises a specialist team of improvement scientists and senior researchers based at King's College London. Its work is funded by King's Health Partners (Guy's and St Thomas' NHS Foundation Trust, King's College Hospital NHS Foundation Trust, King's College London and South London and Maudsley NHS Foundation Trust), Guy's and St Thomas' Charity, the Maudsley Charity and the Health Foundation. NS is the director of London Training & Safety Solutions, which delivers team assessment and training to hospitals on a consultancy basis. No funding source had any role in the design and conduct of the study; collection, management, analysis or interpretation of the data or preparation, review or approval of the manuscript. The views expressed are those of the authors and not necessarily those of the National Health Service, the NIHR or the Department of Health.

**Competing interests** NS is the director of London Training & Safety Solutions, which delivers team assessment and training to hospitals on a consultancy basis.

**Provenance and peer review** Not commissioned; externally peer reviewed.

**Data sharing statement** The qualitative data cannot be shared without identifying participants.

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
