## [Reviewer comments · BMJ Open]

ARTICLE DETAILS

TITLE (PROVISIONAL)	Translating concerns into action: a detailed qualitative evaluation of an interdisciplinary intervention on medical wards
AUTHORS	Pannick, Samuel; Archer, Stephanie; Johnston, Maximillian; Beveridge, Iain; Long, Susannah; Athanasiou, Thanos; Sevdalis, Nick

VERSION 1 - REVIEW

REVIEWER	Graham Martin University of Leicester, UK
REVIEW RETURNED	05-Oct-2016

GENERAL COMMENTS	This paper provides a useful and engagingly written account of a novel intervention—Prospective Clinical Team Surveillance (PCTS)—designed to improve identification of challenges to patient safety, work to address them, and efforts to escalate them to senior managers as necessary. It builds on a cluster randomised trial, which appears to have shown significant impact on outcomes, though as far as I could tell, this has been published only as an abstract to date, rather than subjected to full peer reviewed journal publication. The paper draws on ethnographic insight from a researcher closely involved in six of the seven intervention wards, and two focus groups with staff of various backgrounds involved in the study. This is a slightly thin sample, but the data are used to good effect (though see comment below), and I do not think it should preclude publication. I thought the description of the way the intervention helped to turn normalised deviance on its head was particularly interesting. All in all, the manuscript gives a well constructed and helpful account of what it was about the intervention that appeared to make it work, and where it might be improved – both of these are important if the intervention is to be replicated elsewhere. I have a number of more substantive points and a few more minor ones, which I provide in turn. Major points: 1. The article achieves a good balance between parsimony and completeness in description of the intervention, and helpfully refers to a published protocol for those who need more detail about exactly what it comprises. However, it would be helpful if it presented a few more things. Firstly, what exactly is *prospective* about PCTS? It seemed to be based on retrospective reflection, albeit on the immediate past rather than on things that happened some time ago. Secondly, in the discussion (but not the introduction) there are a few references to similar interventions in the US and Canada, involving analysis from non-clinical colleagues / observers, on which the intervention seemed in part to be based (refs 3, 46, 47). More detail
---

on these, and on their relationship to PCTS, up front would be helpful. Finally, while PCTS does indeed seem to be distinctive, the paper would be enhanced if the authors related it to, and compared it with, other similar interventions that seek to improve proactive identification of safety issues, and the relationship between the 'sharp' and 'blunt' ends of care. Two interventions that come to mind in particular are safety huddles (see e.g. Goldenhar et al., *BMJ Qual Saf* 2013;22:899-906) and patient safety walkrounds (see e.g. my own work, Martin et al., *Joint Commission Journal* 2014;40:303-10). Some of the findings are very much aligned with studies of these similar interventions (e.g. on the "negotiated, acceptable accounts" (page 23 line 25) that they create, rather than an unproblematic window on 'the truth'), and so engaging directly with this literature would both strengthen the contribution and help to make it clear exactly what is distinctive about PCTS.

2. As noted above, the article relies on a relatively small dataset, though to good effect. It is very noticeable in the findings section, however, that the primary source is the two focus groups – I saw only one explicit quotation from the field notes. I think this should be rebalanced, with greater use of fieldnote data, either supporting or replacing at least a few of the references to the focus group data.

3. Page 14 provides an interesting analysis of the way PCTS enabled safety issues to be addressed more promptly than the existing incident reporting system, but exactly *how* it did this was not clear to me. Was this to do with personal contact between the sharp end and the blunt end? Or an opportunity to explain the issues and their seriousness/urgency in a way that was precluded by the limitations of the incident reporting system? Or a means of jointly prioritising what should be done? Or what?

Minor points:

- Page 6 line 33: 'methodologies' should probably read 'methods'
- Page 7 lines 9-34 (et passim): I would avoid the term 'auto-ethnography', as this has been used elsewhere to describe a much more introspective, personally oriented approach than described here, and thus the term might cause confusion. The description of embedded research is clear in showing what the study involved.
- Page 7 line 38 to page 8 line 40: I was not able to glean from this description how many intervention sites were involved in the focus group component of the study – presumably they came from the same hospital as the embedded researcher worked in, but did they come from all six wards, some, or just one?
- Page 9 line 55: two references incorrectly given in Harvard format. One of them appears not to have been included in the reference list.
- Page 10 line 27: ECGS would read better as ECGs.
- Page 20 line 3: 'staff' might read better as 'staff numbers' to avoid ambiguity
- Page 22 lines 33-40: I'm not entirely convinced by this defence – given the active involvement of the ethnographer in the intervention itself, social desirability bias may equally apply to the fieldnotes
- Page 22 line 49: I'm not clear what is meant here by "purely objective" (it is made much clearer in the conclusion, page 23 lines 20-22, so perhaps consider adapting this wording)
- Page 23 lines 12-16: I think this sentence needs to be rephrased given that impact on outcomes is not something that this paper demonstrates and, as noted above, the evidence for this impact appears to come from a published abstract rather than a fully peer reviewed paper, for now.

--	--

REVIEWER	Kristina Schildmeijer Institution of Health and Caring Sciences Linnaeus University Kalmar Sweden
REVIEW RETURNED	09-Nov-2016

GENERAL COMMENTS	Thank you for an interesting paper. My comments; Your article would gain by being shortened. BMJ Open suggests 4000 words and you have 5584. The programme lead was embedded researcher at the institution contributing six of seven study wards. What about the seventh? Two experienced qualitative researchers facilitated the Focus groups. Which were they? Two researchers read and re-read the source material, which were they? You used an inductive theory-generating approach. Krueger and Casey have an analysis method especially made for Focus groups. Which were your arguments when you chose the method of analysis? The six first rows in your result belongs to method. I have never seen references in the result (the result is yours!) and in addition you have both numbers and names of your references in the result text. You have too many quotes while the rest of the text is too limited. It gives a fragmented result which is difficult to read and give no sense of the whole. Page 12 You write eighth with text and 10 in numbers. Usually you text numbers up to 12. Page 13 The quote; You could go down and you could see people...I can't see the connection to Rapid resolution and meaningful managerial follow-up. Page 17 The quote; Post-intervention antibiotics had not... I can't see the connection to Changing teams and the organisation.
--

REVIEWER	Dr Rachel Evley University of Nottingham UK
REVIEW RETURNED	22-Nov-2016

GENERAL COMMENTS	This is a very interesting and novel study with a high degree of relevance to healthcare practice and service improvement. Well-written, organized logically and the author followed the procedures of qualitative approach. Limitations have been addressed. It would be interesting to see if the quotes were from the focus group or interview as my only recommendation for change.
--

VERSION 1 – AUTHOR RESPONSE

Reviewer 1 comments:

1. *What exactly is 'prospective' about PCTS? It seemed to be based on retrospective reflection, albeit on the immediate past.*

We now make clear the two elements of PCTS that make it a prospective, rather than retrospective, system for action.

It is 'prospective' in two respects. First, it proactively seeks out staff concerns, rather than waiting for staff to volunteer them. Second, it can identify care deficiencies that have not yet led to patient harm.

1. *In the discussion (but not the introduction) there are a few references to similar interventions in the US and Canada, on which the intervention seemed in part to be based. More detail on these, and on their relationship to PCTS, up front would be helpful.*

In the Introduction, we now dedicate a paragraph to explaining prospective clinical surveillance and presenting PCTS as a novel extension of those systems:

'Prospective clinical surveillance' is one mechanism for improving care delivery from a frontline perspective. Embedded observers (or visiting facilitators) work with frontline staff to record their experiences of care delivery and its consequences, with a structure to support data capture. Documenting potentially flawed care processes and adverse events, prospective clinical surveillance can produce detailed, unit-level performance assessments in near-real time. It is 'prospective' in two respects. First, it proactively seeks out staff concerns, rather than waiting for staff to volunteer them. Second, it can identify care deficiencies that have not yet led to patient harm. However, its effectiveness as an improvement strategy is unclear. Whether prospective clinical surveillance translates staff concerns into tangible action, and how exactly it might do so, have not previously been explored. Here, we investigate the qualitative impact of prospective clinical *team* surveillance (PCTS), a novel extension of this technique.

2. *The paper would be enhanced if the authors compared it with other interventions that seek to improve proactive identification of safety issues... e.g. patient safety walkrounds. Some of the findings are very much aligned with studies of these similar interventions, creating negotiated accounts rather than an unproblematic window on the truth.*

We have incorporated a more explicit section on this in our Discussion, to broaden the references to the existing literature, and highlight the contribution of PCTS.

Thus, whilst senior executives are encouraged to obtain 'first-hand knowledge of the system', there is no single unproblematic version of the reality of frontline care – however we seek to understand it. One study of executive walkrounds (visits to the 'shop floor') suggested that these visits, whether announced or unexpected, necessarily revealed partial, biased accounts of what was going on. Here too, we found that our structured programme to characterise frontline care was subject to interdisciplinary (and clinico-managerial) disputes and tensions. Prospective clinical surveillance has been described as 'the future of measuring patient safety'. We would now

argue that these systems are as value-laden and subjective as any other attempt to understand frontline care delivery.

3. *Fieldnote data should support or replace at least a few of the references to the focus group data.*

We acknowledge that the fieldnote data was used to inform the analysis and results section, but provided too little direct evidence. We have redressed this balance by using additional direct quotes from the fieldnotes.

'We have two chances of getting [this issue resolved] – fat chance and no chance! We are monitoring [this] on a daily basis but [...] the situation is far from ideal and we have escalated up the chain – not that it has helped.' (Support service manager – Fieldnotes)

'I would like to get some feedback on [these issues] first [...] Currently they are only perception[s], and following investigation it may be that the risk changes, or the issue is solely around communication of processes already in place.' (General manager – Fieldnotes)

'Antibiotics had not been given [...] The junior doctors had not seen the procedure report specifying they were needed. No incident report was done, even though [the registrar] specifically asked if one should be completed.' (Fieldnotes)

'PCTS findings informed the discussion in the Trust morbidity and mortality meeting, chaired by the chief executive, highlighting areas which needed faster examination and transformation, and bringing genuinely new information... Many of these concerns had not been formally addressed either in incident reports or top-down initiatives... It also highlighted new targets for improvement.' (Fieldnotes)

4. *How exactly did PCTS enable safety issues to be addressed more promptly than the existing incident reporting systems?*

We had intended this to be clear in the original manuscript; we have rewritten the relevant section for this purpose.

PCTS therefore brought about faster resolution of safety and quality issues through a combination of mechanisms. First, increasing self-monitoring within ward teams helped anticipate and mitigate potential problems. Second, senior ward staff at the daily briefings were more quickly aware of issues they could resolve, without having to trawl through a backlog of outstanding incident reports. Third, facilitation drew out thematic problems across departments, bringing them to the attention of middle managers who had struggled to understand the difficulties experienced by their staff. Fourth, the clinical impact of these problems was made clearer to senior executives, who were motivated by these novel frontline narratives to pursue change.

5. *Methodologies should read 'methods'; ECGS should be ECGs; staff should be 'staff numbers'.*

These have been changed.

6. *The term auto-ethnography might cause confusion. The description of embedded research is clear in showing what the study involved.*

We have chosen to keep the description of auto-ethnography, as this is a faithful replication of the method used by other authors in similar studies. For example, the references given (e.g., McMullen et al. *Trials* 2015;**16**:242) describe a very similar approach with the researcher both coordinating the study and recording the interactions within it. We believe this complements the description of 'embedded research', whereby the researcher is employed by the host organisation and maintains academic links. The two descriptors together give the reader a better understanding of the methods involved.

7. *How many intervention sites were involved in the focus group study?*

We have clarified this in the text, both in the methods and in the results.

Ethnography and focus groups are recommended for this type of investigation: they are used here to evaluate the intervention at the hospital site most heavily involved in the primary study.

Participants represented each of the clinical divisions that had taken part in the study at this hospital (care of the elderly, acute medicine, respiratory medicine, and gastroenterology).

8. *Two references given incorrectly in Harvard format.*

As suggested by the other reviewer, we have removed these references from the Results section.

9. *Social desirability bias may equally apply to the fieldnotes as the ethnography.*

We have removed the argument that we were able to largely mitigate for social desirability bias.

Our results may also have been influenced by social desirability bias – participants giving the acceptable 'right' answers rather than revealing their true thoughts – and may not generalise fully to other settings.

10. *I'm not sure what is meant here by 'purely objective'. It is made clearer in the Conclusion.*

We expand on our meaning earlier in the Results, to better match the Conclusion (see point 3).

11. *The first sentence of the Conclusion needs to be rephrased, as this paper does not does not demonstrate an impact on outcomes.*

We have removed this phrase and ground the Conclusion more firmly in the findings of this paper.

Healthcare systems can only improve with a detailed operational understanding of how care actually takes place. Prospective clinical surveillance strategies offer a route to this understanding. These strategies are not merely novel measurement tools, as they have been described – even if they do get closer to the realities of frontline care than other safety systems. They still produce negotiated, acceptable accounts, subject to the many interdisciplinary tensions that characterise ward work.

Reviewer 2 comments:

12. *Your article would gain by being shortened.*

We have tried to balance the additional explanations and text required by the reviewers with brevity. The paper's length has been reduced (by approximately 700 words).

13. *The programme lead was embedded researcher at the institution contributing six of the seven wards.*

We have now made clear that the purpose of this paper was to explore the intervention at the site most heavily involved in the study (see point 8).

14. *Researchers facilitated the focus groups, and some analysed the source material. Who were they?*

The staff involved in each section of the analysis (and those who provided material support for the focus groups) are now named.

Focus groups lasted approximately two hours, facilitated by experienced qualitative researchers (N.S., M.J.J, S.A.; also Dr Louise Hull and Ms Tayana Soukup).

Two researchers (S.P., S.A.) read and re-read the source material, adopting an inductive (theory-generating) approach.

15. *What were your arguments for this method of analysis?*

We now make a clearer argument for our choice of analytical technique.

This type of analysis is a flexible research tool, generating a rich and detailed account of a complex data set. It can be applied to focus groups as well as other qualitative data, allowing thematic integration in a single analysis.

16. *The first six rows of your result belong to method.*

Our Methods section describes the mechanisms of data collection; our Results section describes the data set that emerged from this. We believe this is an acceptable convention.

17. *You have too many quotes while the rest of the text is limited. It gives a fragmented result.*

We have tried to balance our revisions to make the Results section more coherent. Some redundant quotes have been removed. At the same time, we have tried to keep the flow of the text from the original version, which the other two reviewers found more engaging than Reviewer 2.

18. *You write eight with text and 10 with numbers.*

We believe this is an acceptable convention. We would be happy to amend this if the Editor feels it necessary.

19. *The quote 'You could go down and see people' – what is the connection to the relevant section (Rapid resolution and meaningful managerial follow-up)?*

We believe this quote is relevant for the start of this section. As we explain in the text, focus group participants drew comparisons with the existing incident reporting system, and its deficiencies, to make clear where they thought PCTS was beneficial. This quote highlighted how poorly the pre-existing system served their needs. We have rephrased the introductory text to better contextualise the quote. The subsequent text then highlights how PCTS was used differently by clinical staff and managers.

Managers too were frustrated by the incident reporting system. Reports did not necessarily illuminate what was happening at the frontline.

20. *The quote 'Post-intervention antibiotics had not' – what is the connection to the relevant section (Changing teams and the organisation)?*

This quote introduces the concept that one possible mechanism for translating PCTS concerns into action – transcribing them first into formal incident reports – was limited. As the remainder of the section explains, PCTS nonetheless had an impact within the team in facilitating incident investigation, and in prompting a more proactive approach to incident management as a whole. Thus there were unit-level and departmental changes as a result of the intervention. The introductory text preceding the quote now makes this more clear.

It provoked new reports, and altered how those reports were addressed. Yet the translation of briefing-recorded incidents into online reports was imperfect, not least because there was no agreed list of high priority events to be recorded.

Reviewer 3:

21. *Were the quotes from the focus group or the interview?*

This has now been made clear (see point 4).

VERSION 2 – REVIEW

REVIEWER	Graham Martin University of Leicester, UK
REVIEW RETURNED	21-Jan-2017

GENERAL COMMENTS	Thank you - you have addressed my comments thoroughly. I look forward to seeing the paper published.
--